# Sex-Specific ADHD-like Behaviour, Altered Metabolic Functions, and Altered EEG Activity in Sialyltransferase ST3GAL5-Deficient Mice

**DOI:** 10.3390/biom11121759

**Published:** 2021-11-24

**Authors:** Tatyana Strekalova, Ekaterina Veniaminova, Evgeniy Svirin, Ekaterina Kopeikina, Tatyana Veremeyko, Amanda W. Y. Yung, Andrey Proshin, Shawn Zheng Kai Tan, Sharafuddin Khairuddin, Lee Wei Lim, Klaus-Peter Lesch, Susanne Walitza, Daniel C. Anthony, Eugene D. Ponomarev

**Affiliations:** 1Department of Psychiatry and Neuropsychology, School for Mental Health and Neuroscience (MHeNS), Maastricht University, 6229 ER Maastricht, The Netherlands; t.strekalova@maastrichtuniversity.nl (T.S.); jogikint@gmail.com (E.S.); kplesch@mail.uni-wuerzburg.de (K.-P.L.); 2Laboratory of Psychiatric Neurobiology, Institute of Molecular Medicine and Department of Normal Physiology, Sechenov First Moscow State Medical University, 119991 Moscow, Russia; katya.veniaminova@gmail.com (E.V.); daniel.anthony@pharm.ox.ac.uk (D.C.A.); 3Laboratory of Cognitive Dysfunctions, Federal State Budgetary Scientific Institution “Institute of General Pathology and Pathophysiology”, 125315 Moscow, Russia; 4School of Biomedical Sciences, Faculty of Medicine, The Chinese University of Hong Kong, Shatin, Hong Kong, China; ekaterina.kopeikina@link.cuhk.edu.hk (E.K.); tatyanaveremeyko@cuhk.edu.hk (T.V.); amandayung@cuhk.edu.hk (A.W.Y.Y.); 5Laboratory of General Physiology of Functional Systems, Federal State Budgetary Scientific Institution, P.K. Anokhin Research Institute of Normal Physiology, 125315 Moscow, Russia; proshin_at@mail.ru; 6School of Biomedical Sciences, Li Ka Shing Faculty of Medicine, The University of Hong Kong, Hong Kong, China; u3004531@connect.hku.hk (S.Z.K.T.); sharafkdin@gmail.com (S.K.); drlimleewei@gmail.com (L.W.L.); 7Division of Molecular Psychiatry, Center of Mental Health, University of Würzburg, 97080 Würzburg, Germany; 8Department for Child and Adolescent Psychiatry and Psychotherapy of the University of Zürich and the Zürich University Hospital of Psychiatry, 8070 Zürich, Switzerland; susanne.walitza@pukzh.ch; 9Department of Pharmacology, Oxford University, OX1 2JD Oxford, UK; 10Kunmin Institute of Zoology, Chinese University of Hong Kong Joint Laboratory of Bioresources and Molecular Research of Common Diseases, Kunming 650000, China

**Keywords:** lactosylceramide alpha-2,3-sialyltransferase (ST3GAL5), attention-deficit/hyperactivity disorder (ADHD), insulin receptor (IR), sex differences, electroencephalogram (EEG), mice

## Abstract

A deficiency in GM3-derived gangliosides, resulting from a lack of lactosylceramide-alpha-2,3-sialyltransferase (ST3GAL5), leads to severe neuropathology, including epilepsy and metabolic abnormalities. Disruption of ganglioside production by this enzyme may also have a role in the development of neuropsychiatric disorders. ST3Gal5 knock-out (*St3gal5^−/−^*) mice lack a-, b-, and c-series gangliosides, but exhibit no overt neuropathology, possibly owing to the production of compensatory 0-series glycosphingolipids. Here, we sought to investigate the possibility that *St3gal5^−/−^* mice might exhibit attention-deficit/hyperactivity disorder (ADHD)-like behaviours. In addition, we evaluated potential metabolic and electroencephalogram (EEG) abnormalities. *St3gal5^−/−^* mice were subjected to behavioural testing, glucose tolerance tests, and the levels of expression of brain and peripheral A and B isoforms of the insulin receptor (IR) were measured. We found that *St3gal5^−/−^* mice exhibit locomotor hyperactivity, impulsivity, neophobia, and anxiety-like behavior. The genotype also altered blood glucose levels and glucose tolerance. A sex bias was consistently found in relation to body mass and peripheral IR expression. Analysis of the EEG revealed an increase in amplitude in *St3gal5^−/−^* mice. Together, *St3gal5^−/−^* mice exhibit ADHD-like behaviours, altered metabolic and EEG measures providing a useful platform for better understanding of the contribution of brain gangliosides to ADHD and associated comorbidities.

## 1. Introduction

Gangliosides are critically involved in protein organization [1,2], signaling, adhesion, and brain development by regulating myelination, insulin receptor function, immunity, and other processes [3,4,5]. Alpha-2,3-sialyltransferase 5 (ST3GAL5) is responsible for the production of GM3, which represents the root structure for 90% of the gangliosides (i.e., GM1, GD1a, GD1b, and GT1b) in the human brain. At the clinic level, biallelic transmission of rare variants of encoding the *ST3GAL5* gene leads to an absence of GM3 synthesis and serious neuropathology [6].

Emerging evidence has refined our understanding of the impact of ganglioside deficiency, which suggests that certain neurodevelopmental psychiatric disorders might be associated with ganglioside dysfunctions. GWAS have reported associations between SNPs encoding ST3GAL3 and the incidence of schizophrenia, attention-deficit/hyperactivity disorder (ADHD) and autistic spectrum disorders (ASD) [7,8]. The ganglioside expression pattern has been proposed as a biomarker of schizophrenia [1,5].

GM3-derived gangliosides are also known to regulate insulin receptor (IR) function, and dysregulation of IR was shown to be associated with increased incidence of ADHD and ASD [9,10]. GM3 regulates IR activation in membrane microdomains by suppressing its phosphorylation [11] and they appear to mediate the inhibitory effects of tumor necrosis factor (TNF) on the IR [12]. In obesity, leptin and insulin resistance are accompanied by increased levels of GM3 synthase in animal and human studies [12,13]; the genetic deletion of ST3GAL5 has rescued insulin resistance in mice fed with a high-fat diet [3].

Ganglioside expression deficiencies in humans have been modelled in *St3gal5^−/−^* mice [3]. While these mutants lack the major CNS gangliosides, they only partially re-capitulate abnormalities observed in the clinic, which may be attributable to the over-expression of the 0-series gangliosides GD1α and GM1b [4]. Thus, *St3gal5^−/−^* mice might be considered as a model of partial deficiency of brain gangliosides associated with developmental psychiatric abnormalities, rather than severe neurological syndromes. Juvenile *St3gal5^−/−^* mice, of both sexes display hyperlocomotion, poor performance in the Y-maze [14], and increased phosphorylation (activation) of the insulin receptor (IR) in skeletal muscle, resilience to high-fat diet-induced insulin resistance [3], reduced platelet activation, and increased neuronal damage following brain trauma [15,16].

However, the behavioural consequences of ganglioside deficiency displayed by the *St3gal5^−/−^* mice have not been investigated extensively in relation to the ADHD-like syndrome. Therefore, we sought to study these features in adult male and female *St3gal5^−/−^* mice. Given previous findings of altered IR function in *St3gal5^−/−^* mice as well as the role of GM3 in the regulation of this receptor, and evidence of altered IR-signaling in ADHD-patients, glucose tolerance and brain, liver and spleen expression of IR isoforms were also examined. As ST3GAL5 dysfunction causes epilepsy and altered electrical cortical activity [5], we also evaluated the electroencephalography (EEG) in the mutant animals.

## 2. Materials and Methods

### 2.1. Animals

*St3gal5^−/−^* and C57BL/6 mice (2–3-months old) were bred [15] and housed under standard conditions; all protocols complied 2010/63/EU, ARRIVE, and Hong Kong Government guidelines (see Appendix A). Independent cohorts were examined for the open field, elevated plus maze, dark/light box, novel object exploration, three-chamber social exploration and tail suspension behaviours, body weight, mRNA levels of IRA and IRB isoforms in the brain cortex and spleen, and brain cortex EEG-parameters (see Appendix A–C). Glucose tolerance test was performed under unchallenged conditions and following saline i.p.-injection challenge. For the protocols employed and group sizes, see SF.

### 2.2. Behavioural Tests

Experiments were performed using customized equipment and analyzed offline (Anymaze, Stoelting, Ireland). Briefly, in the open-field test, distance travelled, speed, and percent of time spent in the central zone were scored; in the elevated-plus maze, mean speed in the central zone, time spent in open arms were recorded [17]. Time spent in the lit box of the dark–light box and the duration of immobility in the tail suspension test were recorded [18]. The duration of the object exploration was measured in the object recognition/exploration test, but owing to the group differences in baseline exploration, object recognition could not be assessed. In the three-chamber sociability test, exploratory rears were also counted; but owing to group differences and the significantly decreased scores in basal exploratory and locomotor activity in the mutants, social interaction could not be examined.

### 2.3. Glucose Tolerance Test (GTT)

The GTT was performed after 18 h of fasting. The mice were gavaged with the glucose solution, and blood glucose concentration were measured at 0, 5, 15, 30, 60, and 120 min [18]. Data were normalized to the baseline values and the area under the curve (AUC) was calculated. This test was performed in naïve mice and in mice subjected to the i.p. saline injection stress 6 h prior to the GTT.

### 2.4. Culling of Mice and Real-Time Reverse Transcription Polymerase Chain Reaction (Real-Time RT-PCR)

Mice were terminally anaesthetized and perfused intracardially with cold PBS; the brain cortex and spleen were removed, homogenized, and stored at −80 °C [15]. Real-time RT-PCR of samples was performed [19] (see Appendix A). Relative gene expression was calculated using the ΔΔCt method and normalized to the expression of the housekeeping gene glyceraldehyde 3-phosphate dehydrogenase (GAPDH) and to the expression of the control sample (see Appendix A).

### 2.5. EEG Study

30-min EEG-recording was performed for four consecutive days on freely moving mice that were stereotactically implanted with electrodes connected to wireless transmitters (BIOPAC Systems, Goleta, CA, USA). The number of high-amplitude peaks (>10 μV) and maximal and minimal peak difference were calculated after data processing with AcqKnowledge 4.4.2 (BIOPAC Systems Inc., Goleta, CA, USA; [16]; Appendix A).

### 2.6. Statistics

Data were checked for normality and treated by two-way ANOVA and Tukey’s tests using GraphPad Prism v.8.01 (San Diego, CA, USA). The level for significance was set at *p* < 0.05.

## 3. Results

### 3.1. St3gal5^−/−^ Mice Display Signs of Hyperactivity, Impulsivity, Anxiety, and Neophobia

A two-way ANOVA revealed that total distance traveled and the speed in the central zone of the open field (*p* < 0.05, Figure 1A,B) was increased in the *St3gal5^−/−^* mice compared with wild-type mice. No differences were found in speed in the open arms of the elevated-plus maze (*p* > 0.05, Figure 1C). The *St3gal5^−/−^* genotype also decreased the time spent in central zone of open field and in the open arms of elevated plus maze (*p* < 0.05, Figure 1D,E), which is suggestive of increased anxiety. There was a sex × genotype interaction for the time spent in lit box of the dark–light box (*p* < 0.05, Figure 1F), which was shorter in *St3gal5^−/−^* females compared to the wild-type females (*p* < 0.05, Tukey’s test) and longer in control females than in wild-type males (*p* < 0.05).

The duration of object exploration exhibited a sex × genotype interaction (*p* < 0.05, Figure 1G), and this measure was decreased in *St3gal5^−/−^* females compared to controls (*p* < 0.05, Tukey’s post-hoc test); there was no difference in the males. There was a significant effect of genotype on the number of exploratory rears in the three-chamber sociability test (*p* < 0.05, Figure 1H), which was decreased in *St3gal5^−/−^* mice in comparison to the controls. A significant genotype effect was also found in the duration of immobility in the tail suspension test (*p* < 0.05, Figure 1H); immobility was decreased in the *St3gal5^−/−^* mice compared to controls (see statistical values in Appendix A).

### 3.2. Metabolic Changes, Expression of Insulin Receptor, and EEG-Parameters in St3gal5^−/−^ Mice

A sex × genotype interaction was significant for mRNA levels of IRA and IRB in the spleen and liver (*p* < 0.05, two-way ANOVA, Figure 2A,B,E,F), but there were no significant interactions for the cortex (Figure 2C,D). In comparison with wild-type males, IRA and IRB expressions were lower in *St3gal5^−/−^* males and wild-type females in both spleen and liver (*p* < 0.05, Tukey’s test; Figure 2A,B,E,F). We found significant sex × genotype interaction in body weight (*p* < 0.05), which was higher in *St3gal5^−/−^* males than controls and female groups (*p* < 0.05, Tukey’s test, Figure 2G). Basal blood glucose levels were unchanged in naïve mice (*p* > 0.05, Figure 2H), but genotype had a significant effect following the i.p.-injection challenge (*p* < 0.05, Figure 2I). There was a significant effect of sex and a sex × genotype interaction on the area under the curve in the naïve and i.p.-challenged mice (*p* < 0.05 and *p* < 0.05; respectively, Figure 2J,K); this parameter was decreased in the challenged *St3gal5^−/−^* mice compared to controls. The *St3gal5^−/−^* genotype also significantly increased the number of high-amplitude EEG-peaks and the mean amplitude of the EEG-peaks (A_max_–A_min_) (*p* < 0.05, Figure 2L–N) compared to controls (see statistical values in Appendix A).

## 4. Discussion

The behavioural findings suggests that the *St3gal5^−/−^* mice are hyperactive, impulsive, neophobic, anxious, and which may be indicative of ADHD-like behaviours. The female mutants also displayed decreased time spent in the anxiogenic areas of the dark–light box, and shortened exploration of novel objects. The hyperactivity, inattention, and impulsivity that are often accompanied by anxiogenic changes are characteristic of the ADHD syndrome [20].

The changes reported here are in keeping with the demonstration of hyperactivity in juvenile male *St3gal5^−/−^* mice bred on C57BL/6-129/sv [3,14] and C57BL/6 [15] backgrounds. The open-field hyperlocomotion of the *St3gal5*^−/−^ mutants was also previously reported in juvenile mice of both sexes [21], which is consistent with our observations here. Niimi et al. [14] found no effect of the administration of psychostimulant methylphenidate on the hyperactivity of juvenile *St3gal5*^−/−^ mice. However, while methylphenidate is widely used in the treatment of ADHD, the treatment response is only reported to be effective in 54% of cases (range 37–75%) [22]. Anxiety-like and neophobic behaviours were increased in both sexes of mutants, though they were more pronounced in females. Such a sexual bias is commonly observed in female ADHD patients [20]. Sex differences have also been reported in *St3gal5^−/−^* mice bred on a C57BL/6-129/sv background [14]. Anxiety disorders are frequently comorbid with ADHD and a higher prevalence of anxiety symptoms in ADHD patients was reported for both children [23,24] and adults [25,26]. Thus, in addition to the hyperactivity and impulsivity, the increases in anxiety-like behavior further supports the view that the behavioural profile of *St3gal5*^−/−^ mice is reminiscent of an ADHD-like syndromic pattern.

The data presented here are also in keeping with previous findings that have suggested a role for gangliosides in the regulation of emotionality: the administration of ganglioside GM1 attenuated alcohol withdrawal-induced anxiety and suppressed novelty exploration in rodents [27,28], and dietary-induced changes in the concentration of brain gangliosides have been shown to be associated with altered exploration in the open field in mice [29].

At the age the mutant behavior was investigated, *St3gal5^−/−^* mice were reported to be deaf [4,30]. Whilst this deafness would impact certain behaviours in mice, it is unlikely to be of relevance in the present study. The measures of anxiety and locomotion have previously been shown to be unaffected by deafness in genetic mouse models [31,32]. Furthermore, the changes in these measures were different in the male and female *St3gal5*^−/−^ mice, which display no sex difference in the development of deafness [4,30].

There was significant effect of genotype and sex on the metabolic parameters studied, which are often altered in patients with ADHD [10]. Male *St3gal5^−/−^* mice exhibited decreased IRA and IRB expressions in the spleen and liver, and increased body weight suggesting dysfunction of IR signaling in the periphery. Oddly, the change in expression were reversed in the *St3gal5*^−/−^ female mice, but there were also no weight changes in the females. The cause of these sex dependent changes remain unclear. Changes in IR signaling in other metabolically active organs (adipose tissue and skeletal muscle) have also been reported for st3gal5-deficient mice earlier, but no sex-specific studies were performed at that time [3]. The expression of IR in the brain was unaltered. Under subtle stress of an i.p. saline injection, genotype had significant effect on glucose tolerance; the *St3gal5^−/−^* mice exhibited a smaller increase in glucose blood levels, which can be caused by acute stress in mice [33]. This finding is in keeping with the literature where the presence of reciprocal interactions between GM3 gangliosides and IR function have been described as well as increases in insulin resistance in the *St3gal5^−/−^* mice challenged with high-fat diet [3]. Given that the dysregulation of IR seems to contribute to neurodevelopmental disorders by multiple pathways [9], our findings warrant further studies to address the inter-relationships between GM3 gangliosides, metabolic dysregulation and ADHD, and, in particular, under challenge conditions.

We also found that genotype affected the number of high-amplitude EEG peaks and the mean amplitude of the EEG peaks. The rationale behind the study of EEG-peak activity in *St3gal5^−/−^* mice was based on the demonstration of disorganized high-amplitude background EEG activity and frequent spike–wave discharges in humans with ST3GAL5 deficiency [5]. Previous studies using the methodology employed here showed elevated frequency of high-amplitude peaks to be associated with increased neuronal electric activity in the cortex during PTZ-induced seizures in mice [16]. Current finding suggests that the changes in the EEG of *St3gal5^−/−^* mutants are similar to those observed in the clinic. As *St3gal5^−/−^* mice showed an increased response to the chemoconvulsants pilocarpine, kainate [21], and pentylenetetrazole [16], these changes in EEG may indicate vulnerability for these chemoconvulsant-triggered seizures.

The response to pentylenetetrazole was also affected by sex. Notably, the targets of pentylenetetrazole are known to be differentially regulated in males and females; distinct male/female alterations in GABA levels may underlie the behavioural sex bias reported here. The changes in EEG-activity may also represent impaired gaiting in the brain in a similar manner to mice lacking St3gal2 or *St3gal3* genes [34] and the finding of increased dendritic spine density in the *St3gal5^−/−^* mutants [15]. Of note, ADHD is frequently comorbid with childhood-onset epilepsy [35], and, vice versa, for children with ADHD, the risk of epileptic seizures was shown to be 2-fold higher than in healthy controls [36].

## 5. Conclusions

The loss of ST3GAL5 leads to neurobiological abnormalities reminiscent of the endophenotypes of developmental neuropsychiatric disorders. *ST3GAL3* gene variants are also associated with a risk for ADHD [7,8], and thus it is clear from our study and others that further exploration of ganglioside deficiency in humans is warranted in larger, sufficiently powered patient cohorts.

## Figures and Tables

**Figure 1 biomolecules-11-01759-f001:**
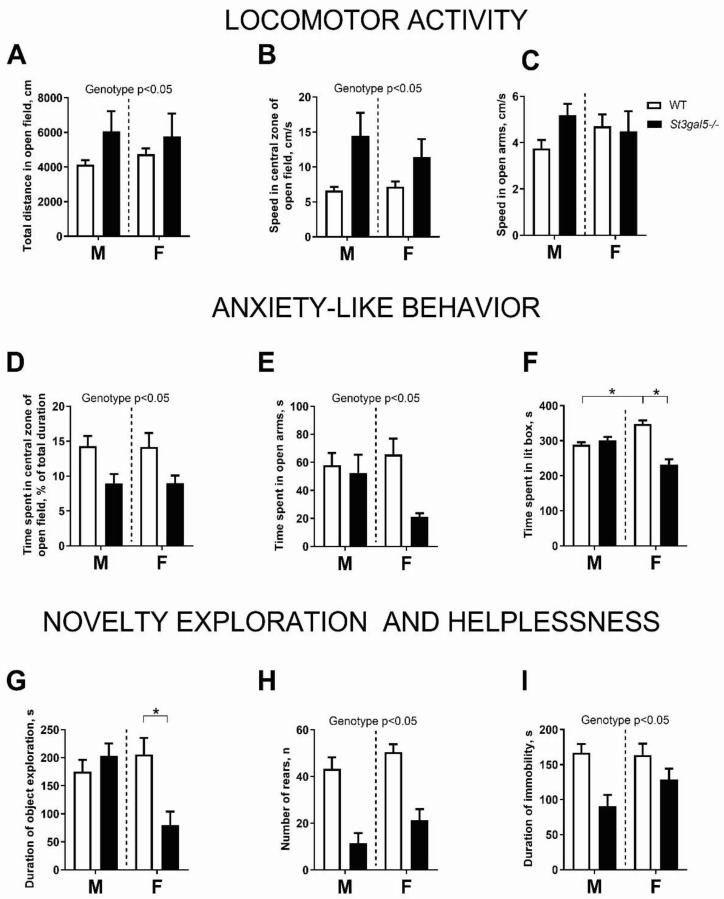
*St3gal5^−/−^* mice display sex bias in behavioural signs of hyperactivity, anxiety, impulsivity and neophobia. Locomotor activity of mice was significantly affected by phenotype for (**A**) total distance moved in the open field, (**B**) mean speed in the central zone of the open field and not (**C**) speed in the open arms of the elevated plus-maze. The anxiety-like behavior of mice was significantly affected by genotype for (**D**) percent of time spent in the central zone of the open field and (**E**) time spent in the open arms of the elevated plus-maze. (**F**) Female *St3gal5^−/−^* mice spent significantly less time in the lit compartment of the dark–light box and (**G**) exploring objects than other groups. There was a significant interaction between genotype and (**H**) the number of rears in the central compartment of three-chamber sociability test and (**I**) the duration of immobility in the tail suspension test. WT—wild-type mice, *St3gal5^−/−^*—*St3gal5^−/−^* mice. * *p* < 0.05 two-way ANOVA and Tukey’s (see text). Data are presented as Mean ± SEM.

**Figure 2 biomolecules-11-01759-f002:**
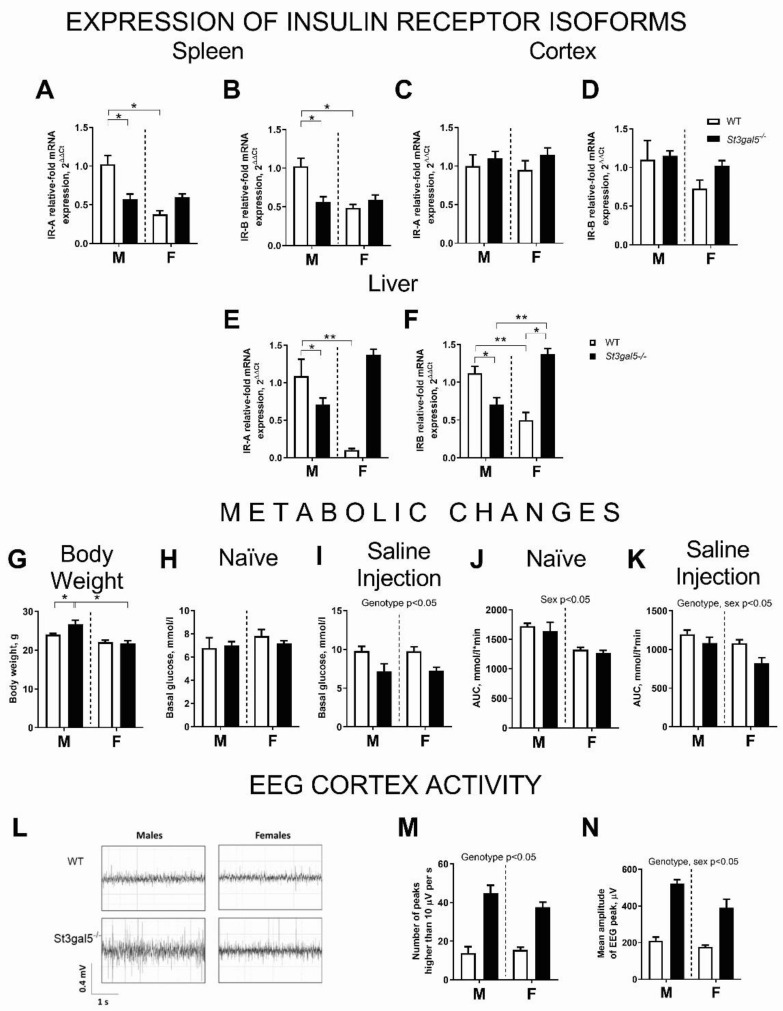
Metabolic changes, altered expression of insulin receptor isoforms, and EEG-parameters in *St3gal5^−/−^* mice. There were significant group changes and a decrease in relative mRNA levels in male *St3gal5^−/−^* mice in spleen (**A**,**B**) and liver (**E**,**F**) expression of (**A**,**E**) IRA and (**B**,**F**) IRB isoforms, but no changes in these measures in the brain cortex (**C**,**D**). Body weight was significantly higher in male *St3gal5^−/−^* mice than in other groups (**G**). Basal glucose levels were (**H**) not altered in naïve mice but (**I**) affected by genotype in i.p.-challenged groups. In this test, sex and genotype affected area under curve in (**J**) naïve mice and (**K**) i.p.-challenged groups, respectively. The representative recordings of EEG-activity in male and female mutants (**L**). There was a significant effect of (**M**) genotype for the number of high amplitude EEG-peaks and (**N**) genotype and sex for the amplitude of EEG-peaks. WT—wild-type mice, *St3gal5^−/−^*—*St3gal5^−/−^* mice. * *p* < 0.05, ** *p* < 0.01, two-way ANOVA and Tukey’s, see the text. Data are presented as Mean ± SEM.

## Data Availability

All the data from this manuscript is available upon reasonable request.

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
