# Peer review of "Sex-Specific ADHD-like Behaviour, Altered Metabolic Functions, and Altered EEG Activity in Sialyltransferase ST3GAL5-Deficient Mice"

_biomolecules, 2021, doi:10.3390/biom11121759_

Round 1
Reviewer 1 Report
The article deals with the specific effect of a- and b- type ganglio-series gangliosides deficiency on complex brain functions, which cannot be rescued by c-type gangliosides.
The topic is interesting and important. The main methodological problem is that st2gal3-/- mice suffer hearing loss and become deaf. The authors should keep this in mind, reporting the levels of deafness at the time of the experiments, and compare them between animals because this may exert a relevant effect on their behavior.
Minor points
Introduction page 2 line 53: use capital letters when referring to the human gene
Page 3 line 120: the digits preceding Ct are not readable
Author Response
For our response to Reviewer 1, please see pages 1-2 and References (pages 9-11) of the attached PDF file.

Reviewer 2 Report
Decision: minor revision
Major comments:
The authors tried to explain the ADHD-like phenotype of St3gal5-/- mice based on their locomotor activity, anxiety-like behavior and novelty exploration/helplessness tests. If the authors describe the ADHD-like phenotype, the following issues should be clarified.
- Figure 1D; The authors describe a decrease in "Time spent in central zone of open field, % of total duration" as ADHD-like symptoms. However, on the contrary, an increase in "Time spent in central zone of open field" is generally a symptom of ADHD animal model. When evaluating the symptoms of ADHD in a pathological animal model, if the symptoms are not a general interpretation, it should be shown that the symptoms are improved by existing therapeutic agent, such as methylphenidate. The authors should explain the above points.
- Figure 1E, 1F and 1I; The authors describe "decreased Time spent in open arms", "decreased Time spent in lit box" and "decreased Duration of immobility(s)"as ADHD-like symptoms. However, these are also not a general interpretation. As I commented as above, the authors should show that these symptoms can be improved by using existing therapeutic agents.
- If the above points are not clarified, the title of this article should be changed from ADHD-like behavior to “abnormal behavior” or “anxiety-related behavior”
- In Discussion:
“Male St3gal5−/− mice exhibited decreased IRA and IRB expression and increased body weight suggesting dysfunction of IR signaling in the periphery, which was not observed in the female mutants. The expression of IR in the brain was unaltered.”
In Figure 2, only the spleen expression of IRA and IRB in WT and St3gal5-/- mice was measured. What about in other major insulin target tissues supposed to be more relevant to metabolism/body weight, such as liver, skeletal muscle, and adipose tissue? Since the data is not suggestive enough as a representation of dysfunctioning peripheral IR signaling, the expression data at least on liver, skeletal muscle, and adipose tissue should be included.
Minor comments:
- Figure 1G; The authors use the "Duration of object exploration (s)" as an index in this figure. However, it is impossible to interpret this result by itself. The author also needs to calculate the percentage of "Duration of object exploration (s)" in the total measurement time. In addition, the percentage of duration to other objects also needs to be considered.
- Figure 1H; The authors evaluate the results only by "Number of rears (n)" in the three-chamber sociability test. However, it is impossible to interpret this result by itself. The authors should use the "Percentage of time spent in each chamber" or "Recognition index".
- Figure 2J, 2K and 2L: The authors evaluate the high amplitude EEG-peaks. Significant differences in genotype are clear. However, I cannot understand the interpretation of the large number of high-amplitude EEG peaks in ADHD animal model. The authors should explain these points.
- References: The authors did not refer the article Tang et al., entitled “Enhanced susceptibility to chemoconvulsant-induced seizures in ganglioside GM3 synthase knockout mice” in ASN Neuro. Jan-Dec 2020;12:1759091420938175 DOI: 10.1177/1759091420938175
Since this article reported a behavioral phenotype of St3gal5-/- mice, the author should cite and discuss about this.
In addition, the authors should reconfirm the accuracy of reference information cited. At least, one reference was not able to find the correct article.
- In Figure 2:“Metabolic changes, altered expression of insulin receptor…” should be rephrased to “Metabolic changes, altered gene expression of insulin receptor isoforms…”.
- Line 68: “ST3GAL3” should be “ST3GAL5”?
- Line 120: ddCT or delta-delta CT?
- Line 137, 139, 143, 158, 161, 163, 170, 172, 177: “St3gal5” should be italicized.
Author Response
For our response to Reviewer 2, please see pages 3-8 and References (pages 9-11) of the attached PDF file.
